# A Method for Mitigating Degradation Effects on Polyamide Textile Yarn During Mechanical Recycling

**DOI:** 10.3390/polym17243243

**Published:** 2025-12-05

**Authors:** Petra Drohsler, Martina Pummerova, Dominika Hanusova, Daniel Sanetrnik, Dagmar Foldynova, Jan Marek, Lenka Martinkova, Vladimir Sedlarik

**Affiliations:** 1Centre of Polymer Systems, University Institute, Tomas Bata University in Zlín, Trida Tomase Bati 5678, 760 01 Zlín, Czech Republic; drohsler@utb.cz (P.D.); d_hanusova@utb.cz (D.H.); dsanetrnik@utb.cz (D.S.); d_sasinkova@utb.cz (D.F.); 2INOTEX, spol. s r.o., Stefanikova 1208, 544 01 Dvur Kralove nad Labem, Czech Republic; marek@inotex.cz (J.M.); martinkova@inotex.cz (L.M.)

**Keywords:** polyamide 66, thermomechanical recycling, chain extender, antioxidant, apparel

## Abstract

The phenomenon of fast fashion has resulted in high yarn consumption and growing textile waste from both manufacturing and consumers. Rising environmental awareness and evolving legislation, including landfill restrictions, have prompted the search for sustainable recycling methods to manage textile end-of-life. This study investigates the mechanical recycling of polyamide 6.6 (PA66) yarn using a chain extender (Joncryl) and antioxidant (Irganox). Thermogravimetric analysis (TGA) confirmed that thermal stability in recycled PA66 was maintained compared to the original yarn, and the presence of Joncryl further enhanced this stability. Oxidative-onset temperature (OOT), measured by differential scanning calorimetry (DSC), supported these improvements. Gas chromatography–mass spectrometry (GC/MS) identified key degradation products, which were correlated with changes in the polymer matrix. Mechanical testing showed a 31% decrease in Young’s modulus after initial recycling, which was reversed with further processing. This behavior suggests the formation of shortened semi-crystalline chains and new linkages promoted by Joncryl. Viscosity and limiting viscosity number increased by up to 50%, depending on both additive concentrations. Overall, Joncryl and Irganox enhanced viscosity, mechanical strength, and notably thermal stability, confirming their suitability for recyclable textile-grade PA66 yarns.

## 1. Introduction

From a scientific perspective, there are numerous approaches to achieving a sustainable lifestyle; however, these methods are often challenging to implement them quickly and effectively under real-world, industrial conditions. Consequently, there is growing interest in reducing material use—not only in the textile industry—as it provides a straightforward and easily applicable solution. Nevertheless, recycling remains one of the key pillars of an environmentally conscious society. Recently, a number of factors have forced society to recycle an increasing proportion of manufactured materials. These factors include the expected shortage of primary resources, limited landfill capacity, government regulations and financial rewards for returning goods for recycling. However, it is equally important to consider textile waste, which represents a significant portion of total waste and often ends up in incinerators or landfills [1]. Recycling offers significant benefits for both the environment and the economy. Unfortunately, it faces many barriers, such as financial, technological, educational, legal, and infrastructural obstacles, among others, which prevent textile recycling from being a successful end-of-life option [2,3].

The rapidly expanding textile industry, driven by fast fashion, consumes substantial quantities of polymeric materials, often derived from non-renewable sources. Globally, approximately 92 million tons of textile waste are generated each year, and this amount is projected to increase to about 134 million tons annually by 2030. This growth contributes to a substantial rise in textile waste, thereby further polluting the environment [4,5,6]. Polyamide 66 (PA66), like polyamide 6, viscose, acrylic, polyester, wool, cotton, polypropylene and polyethylene’s (LDPE and HDPE), belongs to the most commonly used materials from which textile fibers are produced [4]. Approximately 70% of textile waste consists of synthetic materials, primarily polyamide and polyester. Currently, less than 1% of the materials used in clothing production are recycled into new apparels [1].

Polyamides are widely deployed polymers for which demand is growing globally, with production projected to reach 10.4 million tons by 2027. PA66 accounts for approximately one-third of global polyamide production, while polyamide yarn represents about 7% of the polymer fibers applied worldwide [4,5]. Tough, strong and durable, such yarn has found numerous industrial applications. Aliphatic PA66 is a type of polyamide designed for use in fibers and engineering materials. Its key properties relevant to the textile industry include high strength and durability, excellent elasticity, tear resistance, low water absorption, and resistance to rot. For example, PA66 is employed in outdoor sportswear, motorcycle apparel, ropes and parachutes [7].

PA fibers are typically processed by melting, involving extrusion of the polymer through a spinneret at temperatures above its melting point, giving rise to the fibrous product. However, several challenges must be addressed for successful fiber production, including thermal decomposition and fiber breakage, which are common issues during the melt spinning process. Research on the re-fiberization of recycled PA materials, which are particularly prone to thermal degradation, remains very limited, with much of the existing research primarily focused on polyester [8,9,10,11,12,13,14].

However, several degradation mechanisms—notably thermal oxidation, hydrolytic depolymerization, synergistic thermomechanical effects and the formation of low molecular weight volatile compounds—are often not reported in the literature, despite their significant influence on the gradual deterioration of PA66 during repeated recycling cycles.

One way of addressing degradation would be to reduce changes in the molecular weight of the material. A more cost-effective approach, however, involves incorporating a commercially available chain-extending additive. In this process, the additive is introduced in small concentrations (~1 wt%) into recycled polymer pellets, facilitating the reconnection of cleaved polymer chains during melt extrusion [13]. Although its primary function is to enhance chain restoration and improve molecular weight, the additive is also valued for its compatibility with various polymer systems, such as polyethylene terephthalate (PET) in polyamide-based blends [15,16].

Another manifestation of thermal stress in PA is the oxidation of the terminal amino group, which leads to a loss of mechanical properties and causes yellowing. Stabilizers are usually added into the matrix to counteract these phenomena, for example, phenols, phosphite esters and copper salts. Irganox is one of the commercially available and commonly used antioxidants. These are primary antioxidants that act as hydrogen donors, preventing the removal of hydrogen from the polymer’s core structure. Antioxidants are widely used in nearly all commercial polymers and can make up between 0.05% and 3% of their weight [8,14].

Tackling the rise in production of fast fashion and consequent textile waste certainly involves reducing consumption and raising consumer awareness of the related issues. Another approach is to increase the recycled contents of synthetic textile products, yet this often leads to a reduction in their quality [4]. The aim of this study is to maximize the use of reprocessed PA66 textile material, to characterize its properties and the effects of repeated processing, while ensuring the highest possible consistency of material performance through the use of currently available additives suitable for industrial application. Furthermore, to present results on the behavior of polyamide yarn during mechanical recycling (thermal oxidation, synergistic thermomechanical effects and the formation of low molecular weight volatile compounds). In this context, our research focuses on optimizing the thermomechanical processing of PA66 textile yarn by supplementing it with two additives known to prevent degradation mechanisms. Description is given as to how multiple recycling cycles affected samples, along with determination of the amounts of the additives required for each cycle. Material properties were evaluated by conducting mechanical, morphological and thermal tests.

## 2. Materials and Methods

### 2.1. Materials

Polyamide yarn (T-582-23 PADh 66, 2 × 78 dtex f68FT full dull mat, S.C. YARNEA S.R.L.) was supplied by Sintex, a.s., Ceska Trebova, Czech Republic. The chain extender Joncryl ADR4468 (2,3-Epoxypropyl methacrylate; BASF SE, Ludwigshafen, Germany) and antioxidant Irganox 1098 (N,N′-(Hexane-1,6-diyl)bis[3-(3,5-di-tert-butyl-4-hydroxyphenyl)propanamide]; also BASF) were kindly provided by Pigmentum, s.r.o., Pernerova, Czech Republic. Sulphuric acid (96%) was purchased from Penta Labs (Prague, Czech Republic). All of the chemicals were of analytical grade and applied without further purification.

### 2.2. Preparation of Samples

PA66 yarn, employed as a textile waste simulant, was selected as the reference material. Prior to extrusion, the fibers were pretreated by washing in distilled water and subsequently dried to remove potential chemical and physical impurities. The yarn was thenmixed with the Joncryl and Irganox additives via a conical, counter-rotating twin-screw extruder (Thermo Scientific HAAKE MiniLab II, Karlsruhe, Germany). The temperature profile and process parameters were set to a processing temperature of 275 °C and a screw speed of 120 rpm, under a state of continuous melt discharge. The extruder chamber had a volume of 7 mL and was equipped with air cooling.

Initially, the recycling process was simulated by reprocessing the PA66 yarn, designated as cycle 1, followed by regranulation, representing the second such phase of recycling (designated as 2). Characterization was performed on the yarn (hereinafter referred to as “yarn”) as well as on the extruded samples (regranulates). The only exception was the tensile tests, which were conducted on monofil fibers obtained from the rheometer (Goettfert RHEO-GRAPH 50 capillary rheometer (Göttfert Werkstoff-Prüfmaschinen GmbH, Buchen, Germany), equipped with 180° flat entrance angle die with diameter (D) of 1 mm, and a length (L) of 20 mm). The designations of sample and additive ratios are presented in Table 1.

### 2.3. Characterization

Changes in the chemical structures of the PA samples during the recycling treatments and the effects of the additives were determined by Fourier transform infrared spectroscopy (FTIR) on a Nicolet iS5 instrument (Thermo Fisher Scientific, Waltham, MA, USA). The spectrometer using the attenuated total reflection (ATR) technique was applied, with the diamond crystal unit set to a resolution of 2 cm^−1^ and a wavenumber range of 4000–400 cm^−1^.

Thermogravimetric analysis (TGA) was conducted to investigate thermal decomposition on Q500 thermogravimetric analyzer (TA Instruments, Wilmington, DE, USA), operated at a heating rate of 10 °C·min^−1^ and a nitrogen flow rate of 100 mL∙min^−1^. The samples were exposed to temperatures 25–600 °C.

The thermal properties of the materials were studied by differential scanning calorimetry (DSC) using a DSC1 STARe system (Mettler Toledo, Columbus, OH, USA). Measurements were performed at a nitrogen flow rate of 50 mL∙min^−1^ at a heating/cooling rate of 10 °C∙min^−1^. The following program was followed: a heating cycle starting at 25 °C and rising to 400 °C, then cooling to 25 °C. The melting (T_m_) and crystallization temperatures (T_c_) of the samples were determined from the resulting curves. The crystalline fraction was calculated via Equation (1):Χ = (∆H_m/_∆H^0^) × 100,(1)
where ΔH_m_ and ΔH^0^ are the specific melting enthalpy of the polymer and fully crystalline PA66 (196 J∙g^−1^) [17], respectively. It should be noted that the amounts of the chain extender and antioxidant were so low that its contribution to ΔH_m_ could be ignored.

Relative viscosity tests of the reference samples (the yarn and specimens without additives) and others containing the additives were performed according to ISO 307:2007 [18], an Ubbelohde viscometer was employed for this purpose. According to this standard, the viscosity number was first calculated. A 250 mg of each sample dissolved in 50 mL of sulphuric acid (purity 96%) at 50 °C were used. The solution was conditioned in a water bath at 25 °C for 20 min prior to analysis. The results were expressed as the means of three measurements, including a blank. The intrinsic viscosity, and subsequently, the viscosity–average molecular weight was calculated from the relative viscosity via the equations provided below [19].Relative viscosity: η_r_ = η/η_0_,(2)Specific viscosity: η_sp_ = η_r_ − 1,(3)Intrinsic viscosity: [η] = √2 [ η_sp_ − ln(η_r_)]/c,(4)
where c is the concentration of polymer in the solution (in g/mL), calculated as the mass of polymer divided by the volume of the solvent.Molar mass, the Mark–Houwink equation: [η] = k [M_v_]^a^,(5)
where *k* is the Mark–Houwink constant, *a* is the Mark–Houwink exponent related to the polymer structure, and *M_v_* is the viscosity-average molecular weight.

Identifying target compounds in the polyamide involved measuring the molecular ions present gas chromatography on apparatus coupled to a single quadrupole mass spectrometer (GC/MS) was performed on a GCMS-QP2010 Ultra device (Shimadzu, Kyoto, Japan) equipped with a fused silica capillary column (Rxi-5ms, 30 m × 0.25 mm × 0.25 µm, Restek, Bellefonte, PA, USA). Helium was used as the carrier gas at a flow rate of 1.10 mL∙min^−1^. The temperature of the column was held at 40 °C for 2 min and then increased to 320 °C at a rate of 20 °C∙min^−1^, a temperature which was maintained for 13 min, respectively. The range of the scan equaled 29–600 (*m*/*z*) at a speed of 1250, and the entire program lasted 29 min in total. Peaks that appeared in the resultant TIC spectra were identified with the help of the NIST11 Spectra Library.

The oxidation stability of the samples, specifically the oxidation—onset temperature (OOT), was determined on the same DSC instrument under a state of controlled temperature in an air atmosphere.

Tensile testing was performed with adherence to ISO 527-1:2019 [20], on an M350-5CT Testometric machine (Rochdale, UK). The samples measured for this purpose were fibers resulting from rheometric instruments. The measurement length was 50 mm in length and approximately 0.25 mm in thickness. The strain rate was set to 100 mm·min^−1^. Mean average values were calculated from 10 independent measurements taken for each sample. The morphology of the samples after mechanical testing was analyzed using NovaNanoSEM 450 (Eindhoven, The Netherlands, FEI Company) scanning electron microscope (SEM) Imaging was performed using the secondary electron (SE) detector with 5 kV acceleration voltage and 3.0 spot size.

## 3. Results

### 3.1. Structural Changes

Fourier transform infrared spectroscopy (FTIR) is a technique widely deployed to detect the structural properties of samples. By applying the same baseline for all specimen graphs and the correct internal reference, it is possible to measure the integral of the band associated with an identified functional group and quantify it. This technique helps to observe how degradation occurs, as long polyamide chains are broken down into shorter oligomers with amine and carboxylic acid end groups. Figure 1 details the FTIR spectra of the yarn and, for illustration, recycled samples after the first (a) and second (b) reprocessing cycle with selected additives. The specimen of yarn shows medium intensity bands at 3298 cm^−1^ and 2936 cm^−1^, corresponding to N–H and C–H stretching vibrations for amino and methine groups, respectively. Distinct peaks are evident at 1534 cm^−1^ (denoting an N–H bond and C–N stretching) and 1632 cm^−1^, linked to a C=O stretching vibration from the carbonyl group, both characteristic of amide bands (I, II). The bands at 1462 cm^−1^ and 1417 cm^−1^ assigned to –CH_2_– (scissor vibration) as well as at 1370 cm^−1^ corresponding to the –CH_3_ group, were also observed. The crystalline phase of PA66 (α) is marked by a characteristic band around 934 cm^−1^. The band located at 734 cm^−1^ is related to the presence of –(CH_2_)_n_– chains, where *n* ≥ 4. Bands at 687 cm^−1^ and 582 cm^−1^ are assigned to torsion of hydrogen bonds I and II amide mode [21,22,23]. In terms of the first recycling cycle, a decrease in the peak was noted, specifically for the PA sample at 1740 cm^−1^. This is probably due to the decrease in ester groups [24].

The second recycling reprocessing procedure revealed an effect exerted on the absorbance response of samples, including the influence of additives. A decrease is seen in 2PA and 2PA/J, these being specimens absent of any additional amount of Joncryl. In contrast, the 2PA/J/I sample demonstrates the best performance, with absorbance values higher than those for the original yarn. Comparing the peak intensities, although this may not provide a definitive conclusion, it can be suggested that the 2PA/2J/2I and 2PA/2J samples exhibit similarity to the yarn sample from this analytical perspective.

### 3.2. Thermal Properties

TGA constitutes a valuable method for determining the thermal stability of polymer systems during the heating process, based on change in mass. Thermal degradation and the mechanism of decomposition, under the influence of heat, comprise crucial aspects in the optimization of process parameters.

The thermal stability of the analyzed samples was evaluated using thermogravimetric data. These data, presented in Table 2, include temperatures corresponding to initial weight loss—T_10_ (temperature for 10% weight loss), T_50_ (temperature for 50% weight loss) and T_max_ (temperature for maximum weight loss). The relative thermal stability of each sample was evaluated by comparing the decomposition temperatures at different percentage weight losses. The higher the values of T_10_, T_50_ and T_max_, the greater the thermal stability of the composites [25]. The thermal properties were further complemented by additional characterization through DSC analysis, the results of which are presented in Table 3.

Thermal stability increased during the first reprocessing cycle of the yarn, potentially as a consequence of an alteration in the crystalline phase through a rise in the content of the crystalline region. This phenomenon was observed in the PA sample, which exhibited thermal degradation at 14 °C (later than at T_10_), as well as delays corresponding to 10 °C and 20 °C for T_50_ and T_max_, respectively, compared to the original yarn. This effect became even more pronounced with the inclusion of additives; for example, such a delay of 22 °C for PA/J for T_10_ [26,27].

Generally, in samples containing Joncryl, crosslinking also takes place during processing, which facilitates the reconnection of degraded chains and enhances thermal stability [13,15]. These properties were largely maintained in the second recycling phase, with the exceptions of 2PA and 2PA/J, which exhibited slight decreases in T_max_ of 3 °C and 6 °C, respectively. The greater content of Joncryl contributed to maintaining such thermal properties across the two reprocessing cycles. Moreover, the Irganox additive applied in the first recycling phase supported thermal stability in the second of them. This continuation in effect eliminated the need for further supplementation of them beyond their initial incorporation.

DSC curves from the first heating ramp, where the processing history is preserved, were chosen for the evaluation of mechanical degradation of the samples. The values in Table 3 show the changes that occurred when thermal and mechanical stress became apparent. In the first and second recycling cycle, there were no changes in the main melting peak (T_m_), when the temperature was similar to that of the yarn (265 °C). However, degradation could be observed in the PA sample, with the formation of a peak at ca 252 °C, suggesting shorter chains had formed during the thermal processing cycle. The Joncryl additive facilitated the binding of these shortened chains, preventing such degradation from taking place, hence the aforementioned peak was insignificant for such samples.

The PA/J/I experienced a drop in enthalpy (ΔH_m_), although it maintained the same temperatures as the original yarn, indicating a decrease and change in crystallinity [28].

The evaluation of the cooling temperature after the first heating in Table 3 shows two peaks of the exothermic crystallization (T_c1_ and T_c2_). For all samples, the main crystallization peak (T_c2_) was around 235 °C whereas the first peak (T_c1_) varies during cooling and arises according to the number of reprocessing cycles and the content of supplemented additives. This peak could relate to the rate and manner of crystallization of PA66 [14].

After the first reprocessing, the PA sample shows an earlier onset of T_c1_ at 157 °C, indicating the formation of nucleation sites or amorphous chain segments, which accelerates the initial crystallization kinetics. Upon further reprocessing, T_c1_ shifts to 172 °C and ΔH_c1_ decreases by approximately 30%, reflecting the reduction in energy associated with the initial crystallization, leading to chain reorganization, less nucleation, which can reduce, e.g., strength and elasticity. In the 2PA/J and 2PA/2J samples, ΔH_c1_ increases by approximately 59% and 16%, respectively, compared to the original PA66 yarn, demonstrating that Joncryl accelerates nucleation and interconnects chains while maintaining stable crystallinity content and thermal stability. In contrast, in the 2PA/J/I and 2PA/2J/2I samples, the main cold crystallization enthalpy ΔH_c2_ decreases, which can be attributed to chain branching or micro-crosslinking. The presence of Irganox stabilizes the polymer chains (the amorphous phase) and alters the balance between nucleation and crystal growth, leading to a reduction in ΔH_c2_.

PA66 is capable of crystallizing the triclinic α-phase and the pseudohexagonal mesophase, wherein melt quenching gives rise to α′-crystals with a non-planar hydrogen bond arrangement. The processing conducted in this study only brought about α-modifications of the crystals [29,30]. A smaller peak appearing during recycling cycles indicates the initiation of degradation to shorter semi-crystalline chains. A decrease in crystallinity was observed for the 2PA/J/I and 2PA/2J/2I samples, the associated assumption being that both additives promoted long chain branching and physical crosslinking, thereby reducing chain regularity and hindering the crystallization of PA66 [31].

### 3.3. Analysis of Viscosity Parameters

This characterization presents the evaluation of molecular weight-related parameters and structural characteristics of PA66 yarn that was subjected to one or two recycling cycles. The analysis focuses on relative viscosity (RV), viscosity number (VN), intrinsic viscosity (η) and viscosity-average molecular weight (M*_v_*), in order to discern the effects of reprocessing cycles and incorporating additives on the molecular integrity and structures of samples. The results of these parameters are summarized in Table 4.

RV and VN were used to monitor changes in flow behavior and reveal alterations in molecular architecture. Intrinsic viscosity was determined to estimate M*_v_* and provide further insight into chain scission or possible chain extension reactions, particularly in the presence of the additives (Joncryl and Irganox).

The recycling resulted in a modest increase in the RV of the samples, by up to 15%, primarily in the presence of Joncryl. It was presumed that during processing, both the formation and reconnection of molecular chains occur, along with an increase in the crystalline phase content. No significant reduction in RV was observed for the samples containing Irganox, although a slight decrease in the crystalline phase was detected. This reduction could have contributed to the deterioration of certain properties, e.g., mechanical performance. Values for the VN went up after the second recycling cycle, suggesting that the molecular weight did not rise proportionally in line with the content of Joncryl alone. Continued processing appears to promote an increase in VN, which can be attributed to post-polycondensation reactions induced by Joncryl. Prolonged exposure to processing conditions—i.e., a combination of time and temperature—may facilitate limited chain re-linking. This outcome was anticipated and aligns with the intended objective. In contrast, the presence of the Irganox additive suppresses this effect, and the VN values remain most similar to those of the virgin yarn, but only when a doubled dosage of the additive is applied. Furthermore, both additives demonstrate a reduction in intrinsic viscosity, which correlates with a lower viscosity-average molecular weight (M*_v_*). This effect may be attributed to the promotion of chain branching, a phenomenon also induced by the Joncryl additive, as evidenced by the reduced crystalline phase content. One of the hypotheses is that PA66 is capable of undergoing limited crosslinking under the influence of temperature, resulting in an increase in molecular weight. The initial yarn sample may already contain a crosslinking agent, which can increase the relative viscosity during processing. To a lesser extent, this may also occur spontaneously. These structural changes subsequently influence the mechanical properties, which are discussed in the following section those of the original yarn [32,33].

**Table 4 polymers-17-03243-t004:** Relative viscosity (RV), viscosity number (VN), intrinsic viscosity (η), and viscosity-average molecular weight (M*_v_*) of the PA66 yarn samples after the first and second recycling cycles.

Samples Designation	RV (−)	VN (mL/g)	[η] (dL/g)	M*_v_* * (g/mol)
Yarn	1.76	136 ± 2	1.11	25,000
PA	1.98	137 ± 1	1.11	25,500
PA/J	1.98	134 ± 1	1.12	27,800
PA/J/I	1.95	135 ± 2	1.14	35,800
2PA	1.96	137 ± 0	1.12	27,500
2PA/J	2.01	146 ± 1	1.20	50,700
2PA/2J	2.09	149 ± 2	1.21	54,800
2PA/J/I	2.09	143 ± 1	1.15	38,100
2PA/2J/2I	2.00	139 ± 1	1.14	36,000

* [η] = *k*M_v_*^a^*, *k* = 3.6 × 10^−4^, *a* = 0.85 [34].

### 3.4. Mass Spectrometry Characterization of the PA66 Pyrolysis Products

This method was selected to analyze degradation products arising through the reprocessing of the textile materials, as it would facilitate qualitative identification and generate semi-quantitative data based on signal intensity, as well as potentially revealing the relative concentration of each compound investigated. Although the relative peak areas of the detected products were not determined herein, analysis focused on comparing the appearances of peaks at consistent retention times (RT), a parameter considered stable and permitting a degree of analytical confidence.

All mass spectra were evaluated manually and compared against entries in a spectral library. Correlation factors for the identified compounds were ≥80% (indicating a high specificity of the library match), and often exceeded the second-best match by more than 5%. Additionally, the results were compared with data from previously published studies addressing similar degradation phenomena [35,36,37].

The spectra exhibit characteristic peaks corresponding to known degradation products of PA66 generated during pyrolysis, including cyclopentanone (1), and 1,8-diazacyclotetradecane-2,7-dione (2), all primary products and their RTs are listed in Appendix A. Additionally, the samples contain appreciable quantities of compounds such as propylene, butene, 1-hexene, hexan-1-amine and octadecanal, as suggested by the relative intensities observed [38].

Reprocessing the yarn (the PA sample) led to a noticeable reduction in the concentration of the primary degradation products. Among these, cyclopentanone was identified as the principal monitored compound of PA66. It originates from the hydrolysis of hemiaminal groups, it is associated with the concomitant release of other species, e.g., carbon dioxide (CO_2_), ammonia (NH_3_) and various aliphatic amines. Cyclopentanone (CP) acts as a key precursor for subsequent degradation reactions and is a compound considered typical for the designation PA66. It interacts with pyridine derivatives through redox and condensation reactions, potentially forming condensed cyclopentanone structures and pyridine-related compounds [35].

It was thought that thermal reprocessing could exert a limited yet beneficial influence on the properties of polyamide materials, including a reduction in degradation products following a single reprocessing cycle (see Appendix A). Investigation showed that the absolute intensity of CP diminished, in conjunction with a rise in the concentrations of other compounds (e.g., octadecanal). The PA/J sample exhibited a degradation profile comparable to that of the original fibers, whereas the PA/J/I sample showed increased levels of degradation products, particularly CP, the primary degradation product. Although the PA/J/I sample had a higher M_v_, it also produced more primary degradation products.

After the second recycling cycle of the PA sample, a theoretical decrease in the intensity of the main degradation product, CP, was observed (Appendix A). A similar trend was observed for the 2PA/J sample, while the 2PA/2J/2I sample exhibited a less pronounced effect. Interestingly, the intensity levels in the 2PA/2J and 2PA/J/I samples were comparable to those observed after the first recycling cycle. This is notable because these samples displayed high RV, VN, and M_v_. One possible explanation is that the samples still contained a sufficient number of long chains with amide groups capable of forming the degradation product [33].

A peak in the RT region at approximately 8.3 min was also observed in all samples containing the Joncryl additive. This peak may correspond to fragments of Joncryl itself, but it could also indicate the formation of new adducts within the PA66 chain, which fragmented at high temperatures. In the 2PA/J/I sample; however, this peak was of lower intensity compared to the other samples. Oligomers or fragments may be trapped within the amorphous regions, resulting in slower release or different degradation behavior.

Although there was an increase in degradation products in some cases, there were no significant changes in thermal stability based on the analyses mentioned above. The viscosity parameters were also not affected. Therefore, if there were significant changes, we would rather monitor changes in the mechanical behavior of the material.

### 3.5. Oxidation Onset Temperature

This is an accelerated test that enables rapid evaluation of the performance of stabilizing additives. The oxidation-onset temperature (OOT) is measured by dynamic scanning calorimetry (DSC) under an oxygen atmosphere at a constant rate, until the exothermic peak of polymer oxidation is reached. OOT is defined as the temperature at the commencement of the exothermic oxidation peak. In this study, OOT served as a parameter for the behavior of the polymers during the recycling cycles and the effects of the additives that mitigate degradation [39].

Figure 2a illustrates that, for the yarn, the first oxidative exothermic peak with an onset occurred at 175 °C, although this instance of degradation was only observed under these specific conditions of DSC. It might have happened through impregnation of the fiber, or the presence of another additive that had not been removed by washing with water, which disappeared following a recycling phase. Samples with and without additives after the first and second recycling cycles are presented in Figure 2b,c, respectively. The effect on the melting points for all samples was reflected by a shift from 9 °C to higher values in both reprocessing cycles. The Irganox supplemented specimen showed an exceptional shift in its OOT. This was 4 °C in the first cycle, rising to 9 °C in the second, but only after the amount of Irganox in the PA matrix had been doubled. Compared to the yarn, shifts in the OOT of specimens ranged from 5 °C to 16 °C. The effect of the phenolic antioxidant was therefore confirmed; however, it was necessary either to reintroduce it into the polymer matrix during each reprocessing cycle or to increase its initial dosage. The enhanced melt stability of PA66 containing Irganox can be attributed to the sterically hindered phenolic structure of the antioxidant, which effective radical-scavenging functionality. By neutralizing alkyl and peroxy radicals formed during thermo-oxidative degradation, Irganox suppresses chain-scission reactions and delays the onset of oxidation. The antioxidant remains active under the elevated processing temperatures typical for PA66. The stabilization of the amorphous phase, which is particularly susceptible to oxidative attack, further contributes to the observed increase in OOT [40].

### 3.6. Mechanical Properties

The mechanical behavior of the specimens was characterized by tensile testing at a relatively low strain rate (Figure 3). Experiments were conducted tests on PA monofilament fibers and those supplemented with additives upon completion of both recycling cycles. Assessment was made of the Young’s modulus, which describes the relative stiffness of a material, and strain at break, which indicates the ability of a plastic material to resist shape changes without cracking.

The initial processing of textile fibers in this case leads to an increase in Young’s modulus and a reduction in polymer elasticity, as expected. This is evident in the PA sample, where changes in the amorphous phase and enhanced crystallinity were observed, along with the rearrangement of smaller, more mobile polymer chains. In the case of the PA/J sample, this phenomenon was suppressed, with the incorporation of longer chains and crosslinking prevailing, as confirmed by the higher molecular weight and viscosity. However, this did not result in a loss of material elasticity. A different behavior was observed for the PA/J/I sample, where a reduction in the crystalline phase may indicate the effect of the antioxidant. In this case, Young’s modulus decreased, while elongation at break increased significantly. This suggests the dominant effect of Irganox, which may increase the free volume between polymer chains [41,42].

The second recycling cycle of the textile sample (2PA) confirms typical degradation behavior, characterized by a decrease in Young’s modulus and an increase in elasticity, resulting from shorter chains and an increase in free volume. In samples containing only Joncryl, partial suppression of this degradation and retention of elasticity were observed.

The presence of both additives in the 2PA/J/I and 2PA/2J/2I formulations increased the Young’s modulus of the recycled yarn at the expense of elasticity, which could be undesirable in subsequent processing steps such as fiber drawing. As noted in the DSC analysis, these samples exhibited a decrease in crystallinity, contrary to what might be expected. This behavior can again be attributed to chain branching induced by Joncryl, with a synergistic effect from the antioxidant, which promotes stabilization of the amorphous phase.

Scanning electron images of the (2PA/2J/2I—lower strain at break) and (PA/J/I—highest strain at break) are presented in Figure 4. A pronounced neck formation is evident in the 2PA/2J/2I sample—left images, indicating a more ductile deformation prior to failure, whereas no necking is observed in the PA/J/I specimen—right images, whose fracture surface remains more brittle.

## 4. Conclusions

Given the growing interest in and significance of recycled materials, particularly textiles, it is imperative to develop effective strategies for reprocessing textile materials while preserving their original performance. This study aimed to mitigate degradation effects caused by the mechanical recycling of PA66 textile yarn and enable its reuse in textile applications. Commercially available additives were applied: a multifunctional chain extender (Joncryl) and a phenolic antioxidant (Irganox). The investigation focused on the influence of the recycling procedure and additive content on the PA66 matrix, with emphasis on selected thermal, chemical, and mechanical properties.

The samples generally exhibited improved thermal stability after reprocessing, suggesting partial post-polycondensation chain reactions induced by thermal and shear stress, along with changes in crystalline phase content. Minor chain crosslinking may have occurred. Joncryl significantly enhanced thermal stability, as evidenced by T_10_ increases of up to 23 °C with repeated additions during recycling. DSC revealed reduced crystallinity in Irganox-containing samples during the second cycle, while relative viscosity remained stable. This indicates structural rearrangement into branched chains, potentially limiting crystallization.

Viscosity-related properties also increased, especially with Joncryl, which effectively enhanced the molecular weight of PA66, as confirmed by GC/MS. In samples like 2PA/2J and 2PA/J/I, significant degradation still occurred, indicating the presence of long chains that later broke down during further processing. The oxidation-onset temperature (OOT) remained stable in most cases, with PA/J/I and 2PA/2J/2I showing increases of up to 9 °C. The 2PA/J a 2PA/2J shows significant effects depending on its concentration. Regarding mechanical properties, Young’s modulus increases by only 6–7, 5%, which may fall within the experimental error range, while the tensile strength reaches values comparable to the original PA66 yarn. In the 2PA sample, Young’s modulus decreases by 11%, but the strain at break increases by nearly 84%. In the 2PA/J/I and 2PA/2J/2I samples, while Young’s modulus remains similar to that of the PA66 yarn, the strain at break decreases by 42% and 57%, respectively.

Joncryl proved to be an effective additive for repeated thermomechanical processing of textile-grade PA66, particularly when added at each recycling stage. Its incorporation improved thermal stability, viscosity, mechanical properties, and molecular weight. Although the combination of Irganox with Joncryl provided only limited benefits for the intrinsic properties of PA66, their synergistic effect would be better evaluated in terms of fiber spinnability during industrial processing.

## Figures and Tables

**Figure 1 polymers-17-03243-f001:**
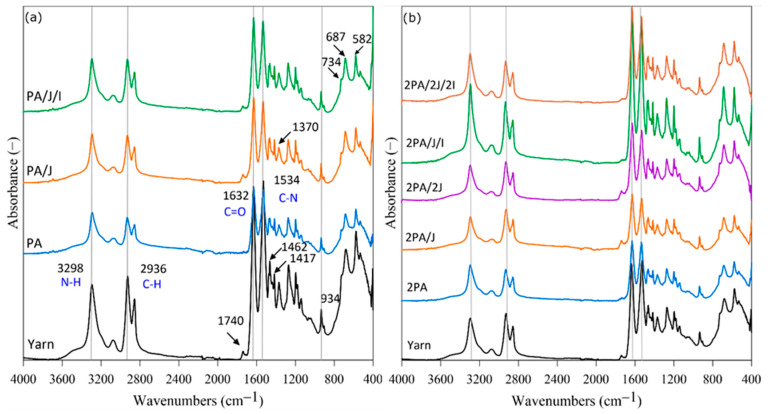
FTIR spectra for the original yarn sample, recycled samples with/without the additives—(**a**) 1st recycling cycle and (**b**) 2nd recycling cycle.

**Figure 2 polymers-17-03243-f002:**
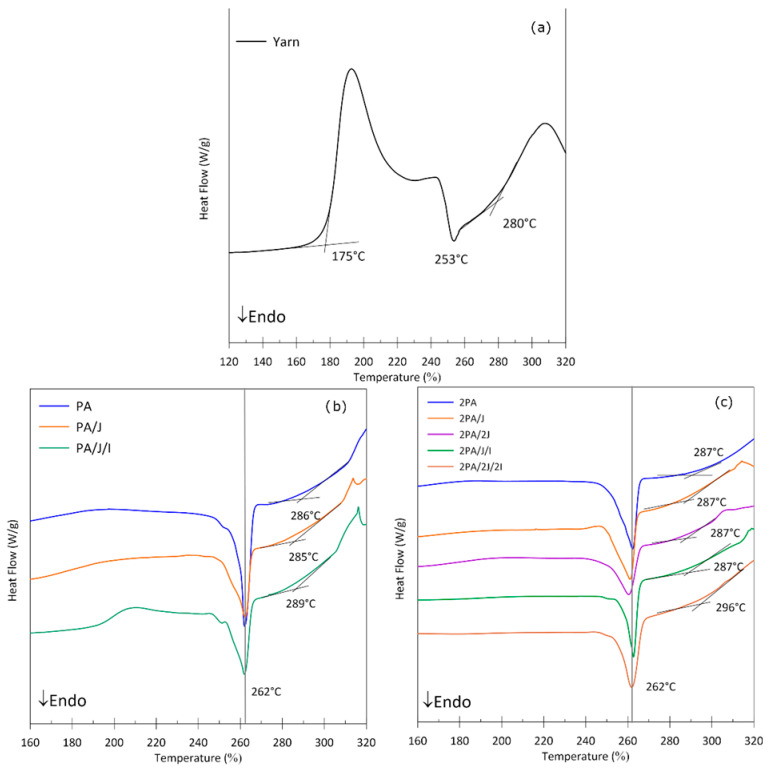
DSC thermograms comparing thermal transitions of (**a**) pristine PA yarn, recycled PA yarn and PA with additives samples after the (**b**) first and (**c**) second recycling cycle.

**Figure 3 polymers-17-03243-f003:**
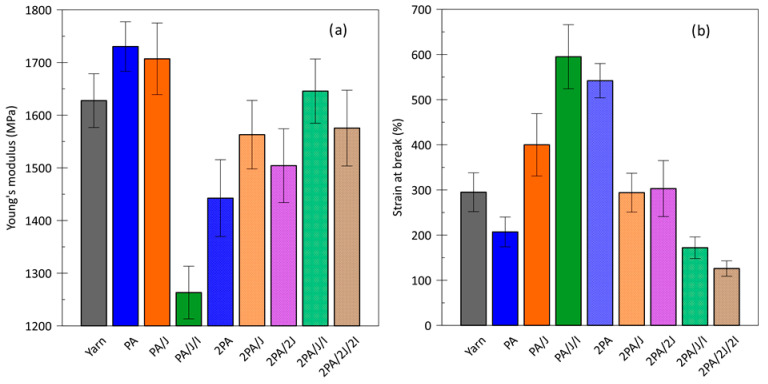
(**a**) Young’s modulus and (**b**) strain at break for the textile yarn and samples with/without the additives as a function of the recycling cycles (*n* = 10; average ± SD).

**Figure 4 polymers-17-03243-f004:**
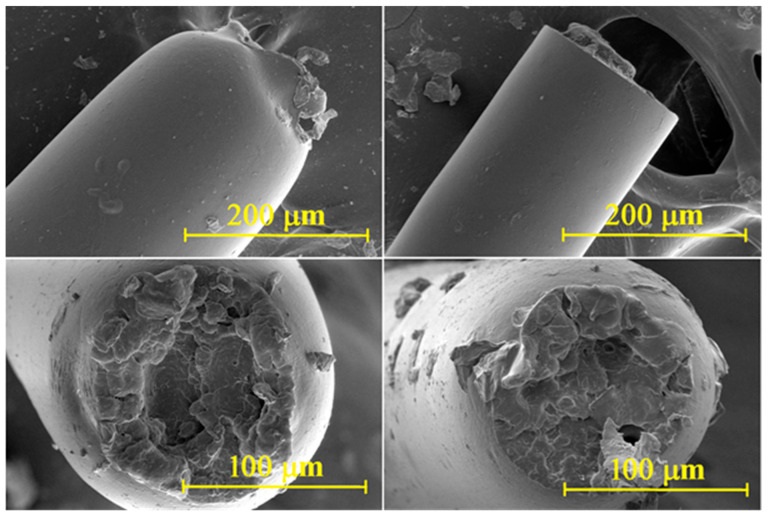
Scanning electron images of the 2PA/2J/2I (left images) and PA/J/I (right images).

**Table 1 polymers-17-03243-t001:** Designation and formulation of the samples.

Input Material *	No. of Processing Cycles	Additive Type	Additive Dosage (%wt.)	Sample Designation
Yarn	1	-	-	PA
Yarn	1	Joncryl	0.5	PA/J
Yarn	1	Joncryl + Irganox	0.5 + 0.1	PA/J/I
PA	2	-	-	2PA
PA/J	2	-	-	2PA/J
PA/J	2	Joncryl	0.5	2PA/2J
PA/J/I	2	-	-	2PA/J/I
PA/J/I	2	Joncryl + Irganox	0.5 + 0.1	2PA/2J/2I

* The initial PA66 yarn, hereinafter referred to as yarn, were subjected to two recycling cycles (1 and 2). The effects of the additives Joncryl and Irganox (denoted as “J” and “I”, respectively), as well as their respective concentrations during each phase of recycling, were systematically investigated. The maximum theoretical concentration of Joncryl was 1%, while that of Irganox was 0.2%.

**Table 2 polymers-17-03243-t002:** Values for T_10_, T_50_ and T_max_ at 600 °C for the PA66 yarn and samples following both recycling cycles, as determined from TGA curves.

Sample Designation	T_10_ (°C)	T_50_ (°C)	T_max_ (°C)
Yarn	372	410	407
PA	386	420	426
PA/J	394	422	423
PA/J/I	390	420	423
2PA	386	418	423
2PA/J	390	418	417
2PA/2J	395	422	423
2PA/J/I	392	421	422
2PA/2J/2I	392	420	422

**Table 3 polymers-17-03243-t003:** DSC data on the PA yarn and PA with additives, after the first and second recycling cycles, alongside values for T_m_, ∆H_m_, X_m_, T_c_, ∆H_c_ and χ gauged from the first heating ramps.

Samples Designation	T_m_ (°C)	ΔH_m_ (J/g)	T_c1_ (°C)	ΔH_c1_ (J/g)	T_c2_ (°C)	ΔH_c2_ (J/g)	χ (%)
Yarn	264	72.0	169	6.1	235	72.5	37
PA	265	77.4	157	5.1	234	70.0	40
PA/J	265	80.3	161	5.3	235	71.3	41
PA/J/I	264	67.0	169	3.3	235	72.6	34
2PA	264	74.8	172	3.6	236	71.9	38
2PA/J	265	71.2	167	9.7	235	70.6	36
2PA/2J	264	73.4	159	7.1	235	71.6	38
2PA/J/I	264	62.0	166	10.7	235	53.1	32
2PA/2J/2I	264	65.8	163	9.1	234	68.6	34

## Data Availability

The raw data supporting the conclusions of this article will be made available by the authors on request.

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
