# Peer review of "Polymers2025, 17(24), 3243;https://doi.org/10.3390/polym17243243"

_polymers, 2025, doi:10.3390/polym17243243_

Round 1

Reviewer 1 Report

Comments and Suggestions for Authors

The manuscript addresses an important and timely issue related to sustainable recycling of textile-grade polyamide (PA66), which is highly relevant to circular economy goals and polymer sustainability research. The authors propose the use of a chain extender (Joncryl) and antioxidant (Irganox) to mitigate the degradation typically observed in mechanically recycled PA66 fibers. The study demonstrates careful experimental planning, integrating thermal (TGA, DSC, OOT), chemical (FTIR, GC/MS), and mechanical characterization to assess the effects of recycling and additive incorporation over two processing cycles. The manuscript is well-structured, scientifically sound, and supported by extensive analytical data. However, several areas require major revision before the publication.

  1. The introduction mentions the environmental impact of PA66 recycling, but does not clearly state the scientific gap. Can the authors specify what specific degradation mechanisms remain insufficiently understood?
  2. How does the chosen PA66 yarn differ from standard industrial-grade PA66 in terms of molecular weight or additives?
  3. The introduction discusses fast fashion but lacks quantitative statistics. Can the authors include global PA66 textile waste data?
  4. Only 10 replicates for tensile testing are mentioned; statistical significance (error bars, standard deviations) should be included for all mechanical data.
  5. What was the melt torque or viscosity profile during extrusion—did the chain extender alter processing rheology?
  6. Was there any evidence of side reactions or gel formation at higher Joncryl concentrations during the second cycle?
  7. FTIR changes are described qualitatively. Did the authors quantify peak-area changes to support claims about chain scission?
  8. DSC cooling curves show double crystallization peaks. Can the authors explain the mechanism behind Tc1 and Tc2 more clearly?
  9. Strain at break is high for PA/J/I. Was necking visually observed? If so, please describe failure mode.
  10. How were Irganox stability and dispersion ensured at 275 °C was any volatilization detected?
  11. Were the observed changes in OOT statistically significant across replicates, or within instrumental error (±1–2 °C)?
  12. Provide microstructural images of the damaged specimens as part of the mechanical properties analysis
  13. The study states that Joncryl is “highly effective,” but this conclusion appears subjective. Can the authors include a quantitative performance metric (e.g., % retention of mechanical properties)?
  14. Do the authors plan to evaluate fiber spinnability or draw ratio performance for these recycled materials in future work?

Author Response

Dear reviewer,

Thank you for your comments and recommendations. We have made changes to the text for its improvement. The Result and discussion part was improved to provide sufficient explanation. The English language has been checked by a native speaker, and in accordance with the recommendation, we have revised the technical terminology and improved the flow of the text.

Yours sincerely, Team of authors

Question:

  1. The introduction mentions the environmental impact of PA66 recycling, but does not clearly state the scientific gap. Can the authors specify what specific degradation mechanisms remain insufficiently understood?

Answer:

Yes, specific degradation mechanisms that were previously not adequately described have now been clarified in the text. The text has been further supplemented and revised to address the scientific gaps that this article aims to fill.

Question:

  1. 2. How does the chosen PA66 yarn differ from standard industrial-grade PA66 in terms of molecular weight or additives?

Answer:

Thank you for your interesting question. Unfortunately, we did not have the opportunity to obtain PA66 granulate as a virgin material. We were only able to characterize PA66 yarns, and during the initial analysis we discovered that the fibers were treated with a non-ionic surfactant (Tween), which adversely affected their processability. Therefore, following standard procedures commonly applied to recycled materials, the yarn was washed (distilled water) and subsequently dried.

Question:

  1. 3. The introduction discusses fast fashion but lacks quantitative statistics. Can the authors include global PA66 textile waste data?

Answer:

Quantitative statistics on fast fashion have been included. However, global data specifically on PA66 textile waste could not be obtained due to insufficient information. No publicly available, reliable statistics exist for the sorting of textile materials by polymer type; therefore, such data could not be incorporated into the article.

Question:

  1. 4. Only 10 replicates for tensile testing are mentioned; statistical significance (error bars, standard deviations) should be included for all mechanical data.

Answer:

According to the relevant standard, which recommend 5–10 replications, we selected the maximum number of test specimens.

Thank you for your the comment; the figure from the tensile tests has been revised to ensure that the standard deviation values are clearly visible for all samples.

Question:

  1. What was the melt torque or viscosity profile during extrusion—did the chain extender alter processing rheology?

Answer:

Thank you for this comment. The online measurement of the melt torque/pressure and viscosity during extrusion are an important parameter for the process evaluation and will be subject to the investigation during the follow-up study, which includes pilot-scale extrusion. In this study, the rheological behaviour was assessed indirectly using RV, VN, η and Mv – in Chapter 3.3. Although torque was not measured, the viscosity data provide a consistent indication of how the additives influenced the melt behaviour. Therefore, direct torque and pressure measurements will be included in the next study.

Question:

  1. Was there any evidence of side reactions or gel formation at higher Joncryl concentrations during the second cycle?

Answer:

Prior to thermomechanical processing, the samples were dried, and the concentration of Joncryl was kept relatively low. Samples containing higher amounts of Joncryl exhibited partially branched chains; however, gelation—which would have led to a reduction in crystallization, viscosity, and elongation—was not observed.

Question:

  1. FTIR changes are described qualitatively. Did the authors quantify peak-area changes to support claims about chain scission?

Answer:

Thank you for your inquiry regarding this analysis.

We calculated the peak area from the OMNIK program and the area content of the C=O/C-H groups. The results were ambiguous and did not indicate the chain length, so we did not include them in the work.

Question:

  1. DSC cooling curves show double crystallization peaks. Can the authors explain the mechanism behind Tc1 and Tc2 more clearly?

Answer:

We appreciate your comment. The text has been revised and expanded to provide a clearer explanation of the differences between Tc1 and Tc2 for the individual samples.

Question:

  1. Strain at break is high for PA/J/I. Was necking visually observed? If so, please describe failure mode.

Answer:

The combination of DSC properties observed in this sample is consistent with its mechanical behaviour. During tensile testing, the elongation of the fibers followed a characteristic pattern: the surface fractured first, followed by the core of the fiber. This indicates that the fiber surface contains fewer crystalline regions due to rapid cooling, making it the first area where cracks initiate.

Question:

  1. How were Irganox stability and dispersion ensured at 275 °C was any volatilization detected?

Answer:

The stability of Irganox was assessed based on the datasheet, and further verified by TGA analysis. The results showed no detectable weight loss of Irganox within the processing window of 275 °C. Its Tonset was observed at temperatures around 351 °C.

No release of Irganox was detected in any of the samples. However, the text in Chapter 3.4 has been revised accordingly. The figures in the supplementary data for this chapter have also been corrected.

Question:

  1. Were the observed changes in OOT statistically significant across replicates, or within instrumental error (±1–2 °C)?

Answer:

Thank you for the comment regarding OOT testing. We based our approach on the literature cited and conducted an additional review of relevant publications, which confirmed that reporting results from a single measurement is commonly practiced.

Question:

  1. Provide microstructural images of the damaged specimens as part of the mechanical properties analysis.

Answer:

Thank you for the suggestion, which would certainly represent a valuable addition to our work. We add the SEM figure (Fig. 4) of samples PA/J/I and 2PA/2J/2I - the ones showing the highest and the lowest elongation at break. These two samples were selected to illustrate the contrast in fracture behaviour

Question:

  1. The study states that Joncryl is “highly effective,” but this conclusion appears subjective. Can the authors include a quantitative performance metric (e.g., % retention of mechanical properties)?

Answer:

We are grateful for your comment. This statement has been corrected in the text.

The sample containing only Joncryl shows, depending on its concentration, notable effects. For instance, in the 2PA/2J sample, thermal stability (T10) increases by up to 6%, and the viscosity-average molecular weight (Mv) increases by 119%. Regarding mechanical properties, Young’s modulus increases by only 6%, which may fall within the experimental error range, while the strain at break reaches comparable values. In the 2PA sample, Young’s modulus decreases by 11%, but the strain at break increases by nearly 84%.

Question:

  1. Do the authors plan to evaluate fiber spinnability or draw ratio performance for these recycled materials in future work?

Answer:

Thank you for your valuable input. This work thus culminates in a spinning process under pilot-plant conditions, where the acquired knowledge is applied. However, the results are proprietary and protected within the project.

Reviewer 2 Report

Comments and Suggestions for Authors

Dear authors,

Thanks for sharing your manuscript. Your paper deals with the thermal reprocessing of Nylon66 fibres and studies the thermal degradation with DSC, IR, tensile properties and volatile formation. Moreover, it tries to comprehend the effects of the addition of chain extender and of a chain extender and an antioxidant to this degradation process. The analytical work is well executed, nonetheless, I would like to discuss several improvement points with you.

First of all, calling thermal reprocessing recycling is slightly condescending for textile recyclers as their encompasses much work; selection of feedstock to have a high concentration of targeted fibres, the removal of untargeted fibres, dirt and other impurities as much as is technically possible and compensating for the presence of the untargeted fibres that cannot be removed. I know this is an academic habit, but that doesn’t make it any better.

Methodologically, I miss a reference sample with only the antioxidant and not the chain extender. This would have helped you to comprehend the effects of the antioxidant and to unravel the actions of the chain extender and the antioxidant.

In general, I find your article mostly descriptive and insufficiently analysing the results of the various techniques in combination. For example, I find that your explanation of why the combined effect of antioxidant and chain extender results in a lower degree of crystallinity in comparison to the single effect of the chain extender insufficient. Moreover, you hardly connect the peculiar impacts on the formation of volatiles as markers of degradation on the thermal properties and mechanical properties, this is a missed opportunity.

Line 45. If recycling would offer benefits for the economy, why are so many recyclers going bankrupt? You miss the geopolitical aspect here, that we are dealing with a large economical power that dumps its products below the market price on the European market to gain control.

Line 88-92. Than you still need the industry to accommodate these feedstocks. If all the production is in Asia than you cannot recycle effectively in Europe.

Line 159, formula 4. You do not explain c in the text.

Line 322 & line 473-474. You mention “chain branching”, did you find gel-formation? Did you find evidence of more complicated spinning the material?

Line 368-370. This is a very interesting observation. Would the antioxidant catalyse the formation of volatiles?

Line 408-410: No the reason is that the antioxidant scavenges radicals.

Line 438-441: Do you really suggest that the antioxidant causes chain branching???

Line 446: We apparently both like recycling very much, and indeed there is a growing academic interest, but not a business interest, as you know, because we are flooded with cheap virgin products.

Good luck

Good luck

Comments on the Quality of English Language

Minor language issues

Quirky sentences that need to be improved: 42-44.

Line 72: “remodel”? To restore perhaps.

Line 77: “reformation”? -> restoration.

Line 116: Again you do not simulate recycling here, you only simulate the thermal reprocessing step of a whole recycling process here.

Line 206-207. Abstruse.

Line 259: “internal enthalpy”???

Line 262-266. Abstruse.

Line 278. What do you imply with “chain disruption”? do you mean “chain scission” or “a conformational twist”?

Line 407: “add it back”-> “replenish” perhaps?

Author Response

Dear reviewer,

Thank you for your comments and recommendations. We have made changes to the text for its improvement. The Result and discussion part was improved to provide sufficient explanation. The English language has been checked by a native speaker, and in accordance with the recommendation, we have revised the technical terminology and improved the flow of the text.

Yours sincerely, Team of authors

Question:

  1. First of all, calling thermal reprocessing recycling is slightly condescending for textile recyclers as their encompasses much work; selection of feedstock to have a high concentration of targeted fibres, the removal of untargeted fibres, dirt and other impurities as much as is technically possible and compensating for the presence of the untargeted fibres that cannot be removed. I know this is an academic habit, but that doesn’t make it any better.

Answer:

Thank you for your interesting comment regarding mechanical recycling.

I would like to clarify that the sample we used was provided as a waste product from primary production. It was removed from the coil and washed, following a commonly applied recycling procedure, and subsequently thermomechanically processed with additives intended to minimize chemical changes and restore the original properties.

Moreover, based on the literature we reviewed, this approach corresponds precisely to the processing route described and applied in our work.

Question:

  1. Methodologically, I miss a reference sample with only the antioxidant and not the chain extender. This would have helped you to comprehend the effects of the antioxidant and to unravel the actions of the chain extender and the antioxidant.

Answer:

Thank you for your the factual comment.

We agree that the work could be more comprehensive, and we plan to address the mechanism of action of the selected antioxidants on this material in greater detail in our future research. However, in this study we also relied on previously published literature in which the effects of this antioxidant had already been investigated.

Question:

  1. In general, I find your article mostly descriptive and insufficiently analysing the results of the various techniques in combination. For example, I find that your explanation of why the combined effect of antioxidant and chain extender results in a lower degree of crystallinity in comparison to the single effect of the chain extender insufficient. Moreover, you hardly connect the peculiar impacts on the formation of volatiles as markers of degradation on the thermal properties and mechanical properties, this is a missed opportunity.

Answer:

Thank you for your the comment.

You are certainly right. The text has now been revised and supplemented accordingly.

Question:

  1. 5. Line 45. If recycling would offer benefits for the economy, why are so many recyclers going bankrupt? You miss the geopolitical aspect here, that we are dealing with a large economical power that dumps its products below the market price on the European market to gain control.

Answer:

I see that we are looking at the issue in a similar way.

However, the article takes into account the possibility of future changes, which may be reflected, for example, in legislative measures or increased fees for the import of these materials. In the future, we will certainly have to take into account the growing carbon footprint associated with the transportation and consumption of minerals.

Question:

  1. 6. Line 88-92. Than you still need the industry to accommodate these feedstocks. If all the production is in Asia than you cannot recycle effectively in Europe.

Answer:

It is important to simplify the mechanical recycling process and utilize low-cost input materials, such as waste materials with zero acquisition cost, to ensure the engagement of recycling companies. This work also highlighted the need to raise consumer awareness – and thus increase interest in recycled textiles.

Nevertheless, it should be emphasized that the scope of this study is not limited exclusively to a European audience.

Question:

  1. 7. Line 159, formula 4. You do not explain c in the text.

Answer:

Thank you for your comment. We have added it to the text.

Question:

  1. 8. Line 322 & line 473-474. You mention “chain branching”, did you find gel-formation? Did you find evidence of more complicated spinning the material?

Answer:

Thank you for your valuable question.

The statement regarding chain branching has also been clarified and further elaborated in Chapter 3.2, in the DSC analysis section, where the results can be correlated with the mechanical properties.

Question:

  1. 9. Line 368-370. This is a very interesting observation. Would the antioxidant catalyse the formation of volatiles?

Answer:

Thank you for your interesting question.

You are referring to the text in chapter 3.4, this chapter has been edited to better describe the results.

Question:

  1. 10. Line 408-410: No, the reason is that the antioxidant scavenges radicals.

Answer:

Question:

  1. Line 438-441: Do you really suggest that the antioxidant causes chain branching???

Answer:

Thank you for pointing out the ambiguity in the text.

The passage has now been clarified and revised accordingly.

Question:

  1. Line 446: We apparently both like recycling very much, and indeed there is a growing academic interest, but not a business interest, as you know, because we are flooded with cheap virgin products.

Answer:

Comments on the Quality of English Language

Answer:

Thank you for your revisions. The text has been edited according to the comments and marked in the text.

Round 2

Reviewer 1 Report

Comments and Suggestions for Authors

Based on the improvements made and the adequacy of the revisions, I recommend acceptance of the manuscript in its current form.